# *Mahonia aquifolium* Extracts Promote Doxorubicin Effects against Lung Adenocarcinoma Cells In Vitro

**DOI:** 10.3390/molecules25225233

**Published:** 2020-11-10

**Authors:** Ana Damjanović, Branka Kolundžija, Ivana Z. Matić, Ana Krivokuća, Gordana Zdunić, Katarina Šavikin, Radmila Janković, Jelena Antić Stanković, Tatjana P. Stanojković

**Affiliations:** 1Department for Experimental Oncology, Institute of Oncology and Radiology of Serbia, 11 000 Belgrade, Serbia; zaanu011@gmail.com (A.D.); branka.kolundzija@gmail.com (B.K.); ivanamatic2103@gmail.com (I.Z.M.); krivokuca.ana@gmail.com (A.K.); jankovicr@ncrc.ac.rs (R.J.); stanojkovict@ncrc.ac.rs (T.P.S.); 2Department for Pharmaceutical Investigations and Development, Institute for Medicinal Plant Research, Dr. Josif Pančić, 11 070 Belgrade, Serbia; gzdunic@mocbilja.rs (G.Z.); katarina.savikin@gmail.com (K.Š.); 3Department for Microbiology and Immunology, Faculty of Pharmacy, University of Belgrade, 11 221 Belgrade, Serbia

**Keywords:** doxorubicin, *Mahonia aquifolium*, matrix metalloproteinases, cytotoxicity, human lung adenocarcinoma

## Abstract

*Mahonia aquifolium* and its secondary metabolites have been shown to have anticancer potential. We performed MTT, scratch, and colony formation assays; analyzed cell cycle phase distribution and doxorubicin uptake and retention with flow cytometry; and detected alterations in the expression of genes involved in the formation of cell–cell interactions and migration using quantitative real-time PCR following treatment of lung adenocarcinoma cells with doxorubicin, *M. aquifolium* extracts, or their combination. MTT assay results suggested strong synergistic effects of the combined treatments, and their application led to an increase in cell numbers in the subG1 phase of the cell cycle. Both extracts were shown to prolong doxorubicin retention time in cancer cells, while the application of doxorubicin/extract combination led to a decrease in *MMP9* expression. Furthermore, cells treated with doxorubicin/extract combinations were shown to have lower migratory and colony formation potentials than untreated cells or cells treated with doxorubicin alone. The obtained results suggest that nontoxic *M. aquifolium* extracts can enhance the activity of doxorubicin, thus potentially allowing the application of lower doxorubicin doses in vivo, which may decrease its toxic effects in normal tissues.

## 1. Introduction

Doxorubicin (DOX) is a first-line anticancer agent that is highly effective against a wide spectrum of malignancies, including breast, lung, gastric, ovarian, and thyroid ones, as well as lymphoma, myeloma, sarcoma, and some forms of pediatric neoplasms. Despite good clinical effectiveness, DOX induces cumulative, dose-dependent toxicity and adverse effects, such as cardiotoxicity, and affects the brain, kidney, and liver [1]. Currently, both cancer treatments and in vivo studies are usually based on combined therapies that include various antineoplastic agents, possibly resulting in drug–drug interactions and even an increase in toxicity [2]. Studies investigating the management of DOX-induced toxicity have focused on the administration of antioxidant and/or antiapoptotic compounds in combination with DOX, the development of effective delivery systems, and the synthesis of DOX analogs [1]. One potential approach to the minimization of adverse effects is reducing the therapeutic dose of DOX by combining its application with that of other anticancer and/or organ-protective agents [3]. However, although some of these strategies fail to decrease DOX toxicity, recent investigations have demonstrated that certain phytocompounds in combination with DOX can ultimately be more successful [4,5].

The genus *Mahonia* includes approximately 60 species, which are widely distributed throughout Asia, North America, and Europe. Species belonging to the genus *Mahonia*, including the *Mahonia aquifolium* plant, have been shown to have antibacterial, antifungal, anti-inflammatory, and antioxidant effects and have been used in traditional Chinese and North American medicine [6]. Some research has shown that several representatives of this genus, such as *Mahonia oiwakensis* and *Mahonia bealei*, which are native to China, demonstrate antiproliferative activity against human cancer cells as well [7,8]. Berberine and similar alkaloids represent a major class of secondary metabolites of the *Mahonia* genus with a wide spectrum of different properties. These alkaloids have been reported to significantly inhibit growth of cancer cells and exhibit other anticancer effects [9,10,11,12]. Although *M. aquifolium* has been used in traditional medicine solely for treatment of inflammatory skin disorders [13], its chemical composition, as well as previously obtained results demonstrating the activity of different plants belonging to this genus, suggest that this plant possesses anticancer properties as well, as we have previously confirmed and reported [14].

Previous studies have demonstrated that the phytocompound berberine in combination with DOX can effectively limit the toxicity and adverse effects of DOX [4] and that *M. aquifolium*, whose main constituents are berberine and protoberberine alkaloids, has anticancer properties [14,15]. Therefore, we investigated the anticancer efficacy of the combination of DOX and water or ethanol extracts of *M. aquifolium* (MAW and MAE, respectively) in vitro.

The objective of our study was to elucidate the effects of DOX and MAW or MAE combinations on proliferation, migratory potential, and invasiveness of malignant cells. Furthermore, we examined the influence of these extracts on cellular uptake and retention of DOX. In order to understand the mechanisms underlying the effects of these extracts on migration and invasiveness, we analyzed gene expression changes of matrix metalloproteinases 2 and 9 (*MMP2* and *MMP9*), occludin (*OCLN*), catenin beta-1 (*CTNNB1*), and excision repair cross-complementation group 1 (*ERCC1*) in the treated human malignant cells.

## 2. Results

### 2.1. Cytotoxic Activity In Vitro

#### 2.1.1. Cytotoxic Activity of Extracts and DOX

MAW and MAE showed moderate cytotoxic activities against A549 cells. After 72 h of treatment with extracts, MAW IC_50_ value was shown to be 56.36 ± 0.30 µg/mL, while MAE IC_50_ was 51.97 ± 3.27 µg/mL. The IC_50_ of DOX was 0.44 ± 0.02 µg/mL.

#### 2.1.2. Cytotoxic Activity of DOX in Combination with Extracts

The combined extracts and DOX effects were evaluated using an isobolographic analysis method. After incubating cells with subtoxic concentrations of extracts in combination with DOX, there was an increase in cytotoxicity compared to the controls (Table 1). The CI values ranged from 0.14 to 0.38 for DOX/MAW and from 0.12 to 0.4 for DOX/MAE treatment, suggesting strong synergism (Table 2).

### 2.2. Cell Cycle Analysis

A549 cells treated with subtoxic concentrations of both MAE and MAW extracts for 24 h showed a slight increase in the percentage of cells in the G2/M phase of the cell cycle, but the changes induced by this treatment did not significantly differ from those in the controls. However, treatment of cells with IC_20_ and IC_50_ DOX led to a significant increase in the percentage of cells in the G2/M phase (Figure 1A), while cells treated with combination of DOX (IC_20_ or IC_50_) and MAW or MAE (20 µg/mL) were shown to accumulate in the G1 phase compared to those treated with DOX alone (Figure 1B). Samples treated with combinations of DOX (IC_20_ or IC_50_) and extracts showed an increase in the subG1 phase as well, compared to samples treated with DOX alone, and this increase in the number of subG1 cells was shown to be statistically significant for cells treated with combination of DOX and MAE (Figure 1B).

### 2.3. Cellular Uptake and DOX Retention

No significant changes in DOX uptake were observed following treatment with extracts (Table 3). However, the effects of extract treatment on DOX retention were more pronounced. Cells treated with extracts after DOX treatment retained considerably more DOX (up to 22% more) than cells treated with DOX and medium only (Table 3).

### 2.4. Cell Migration

Both extracts demonstrated an improved ability to reduce cell migration compared to the control or DOX (Figure 2). After 48 h, this decrease was shown to be statistically significant in both extract-treated samples. Cells treated with DOX alone did not show any significant variation, while the migratory ability of cells treated with DOX and MAE or MAW was shown to be considerably lower after 48 h (Figure 2B).

### 2.5. Colony Formation

As shown in Figure 3, DOX considerably affected the ability of cells to form colonies even when applied in subtoxic concentrations, unlike MAW and MAE extracts, which caused only a slight decrease in the colony-forming ability. However, combinations of DOX and MAE/MAW extracts were shown to lead to an even more pronounced decrease in the colony-forming ability of the treated cells compared to DOX treatment alone.

### 2.6. Gene Expression Analyses

We investigated the gene expression involved in the formation of cell junctions, in cell migration, and in DNA repair, as well as whether this gene expression is associated with the metastatic potential of cells (*MMP2*, *MMP9*, *OCLN*, *CTNNB1*, and *ERCC1*). Gene expression levels in the treated A549 cells were compared with those measured in the untreated, control cells grown only in the nutrient medium (Figure 4).

DOX treatment was shown to decrease *MMP2* expression, while both the applied extracts and DOX/extract combinations slightly increased the expression of this gene (Figure 4). In contrast to this, *MMP9* expression was considerably lower after treatment with combination of the cytostatic and the investigated plant extracts (Figure 4), but it increased after treatment with DOX alone, in contrast to the control. Furthermore, DOX treatment was shown to inhibit the expression of *OCLN* and induce *CTNNB1* expression. Both the investigated extracts alone and the DOX/MAE combination treatment induced expression of *OCLN* (Figure 4) as opposed to the control cells, while its expression in cells treated with the DOX/MAW combination was higher than in cells treated with DOX alone. There were no alterations in *CTNNB1* expression after treatment with the DOX/MAW combination, while its expression was shown to be slightly lower after treatment of cells with the DOX/MAE combination than in the untreated cells (Figure 4).

Cells treated with MAE alone showed a slight increase in *ERCC1* expression levels, while inhibition of this gene expression was observed in samples treated with DOX, in those treated with MAW, as well as in those treated with the DOX/plant extract combination (Figure 4).

## 3. Discussion

Despite decades of good results in the clinical application of DOX in cancer therapy, this drug induces cumulative, dose-dependent adverse effects. Our previous studies have demonstrated that ethanol and water extracts obtained from *M. aquifolium* show good anticancer potential and that berberine and berberine-type alkaloids can be detected in both extracts [14]. Higher content of berberine was detected in MAE extract (2.44%) compared to MAW extract (1.34%) (LC-MC analyses).

Furthermore, after identifying cytotoxic metabolites from *M. aquifolium* using ^1^H NMR-based metabolomic approach, we concluded that alkaloids with the highest cytotoxicity in our extracts are berberine, palmatine, and the bisbenzylisoquinoline alkaloid berbamine [15]. It has previously been reported that berberine and similar alkaloids can inhibit the growth of cancer cells [9], effectively limiting the toxicity of DOX [4].

Our initial screenings demonstrated that, of the tested cell lines, A549 cells were the least sensitive to the cytotoxic activity of MAE and MAW [14]. Nonsmall cell lung cancer patients often show resistance to therapy [16], and several mechanisms underlying the development of multidrug resistance in lung cancer have been identified, such as overexpression of drug efflux proteins and ATP-binding cassette (ABC) transporters [17]. Many studies have confirmed the presence of ABC transporters, breast cancer-resistant protein (BCRP), and lung resistance-related protein (LRP) in A549 cells, which have been shown to be related to anticancer drug resistance [18,19,20]; therefore, we selected the A549 cell line for all further experiments. The results obtained here demonstrate that the IC_50_ concentration of DOX can be multiply reduced (19 to 123 times) when DOX is combined with *M. aquifolium* extracts, suggesting that the same antiproliferative effects can be achieved using much lower concentrations of this drug. Furthermore, DOX and the plant extracts showed strong synergistic effects, clearly demonstrating that the individual anticancer activities of both constituents were preserved. It has been reported that cardiomyophaty, the most important adverse effect of DOX, primarily depends on the applied dose [21]. Doses below 450 mg/m^2^ reduce the frequency of on-treatment events, but the cumulative effects lead to the development of late-onset adverse events [22,23]. Based on this, we hereby propose that coadministration of DOX with an additional agent with a synergistic effect may decrease the toxicity of this treatment without affecting the anticancer activity of DOX. Our previous studies have shown that *M. aquifolium* extracts are several times less cytotoxic to normal, healthy cells than to cancer cells in vitro [14], indicating good selectivity and potential for their use in anticancer therapy.

We conducted the experiments on A549 cells, which are the most invasive but also the least sensitive to DOX, which is part of medical therapy in the treatment of lung cancer. Previous research has shown that both berberine and berbamine can inhibit the growth of lung cancer cells in in vivo systems [24,25], and based on these results, we can conclude that the active principles of our extracts have potential to reach this target in the body.

A549 cell cycle analysis has demonstrated that DOX induces a strong G2/M transition block [26] as well as a considerable increase in the percentage of cells in the subG1 phase in contrast to that in the control sample, indicating that treated cells cannot pass through mitosis, which ultimately leads to apoptosis [27,28]. We demonstrated that the percentage of cells in the subG1 phase was similar when cells were treated with IC_50_ DOX alone or with plant extracts and IC_20_ DOX combined, confirming that the extracts allowed maintenance of the anticancer activity of DOX even when applying lower drug doses. Furthermore, we observed an increase in the number of cells in the G1 phase following treatment of cells with DOX/plant extract combinations in contrast to the number of cells in the G1 phase treated with DOX alone. As one of the goals of drug discovery efforts today is identifying the agents that target cell cycle checkpoints responsible for the control of progression through the cell cycle [29], we believe that the results obtained may be very important. Cell percentage increase in the G1 and subG1 phases observed in samples treated with DOX/plant extract combinations may suggest that this G1 block is irreversible and that treatment induces apoptosis, which leads to an increase in the number of cells in the subG1 phase. This significantly higher percentage in samples treated with the DOX/MAE combination than in the control sample, as well as the existing DOX-induced G2/M arrest observed after the treatment of cells with both MAE/MAW and DOX combinations, additionally confirms their synergistic effects.

Although extracts did not affect DOX uptake, we have demonstrated that they induce the retention of DOX up to 20% more than in untreated samples. Increasing drug dose to overcome drug resistance in cancer therapy is not feasible due to numerous potential side effects [30], and alternative approaches include improving accumulation, prolonging retention of drugs in cancer cells [31], and reducing drug exposure time [32]. The inhibition of drug efflux transporters p-glycoprotein (Pgp) and BCRP restores the intracellular levels of drug in DOX-resistant osteosarcoma cells and leads to the retention of DOX [33]. Our research suggests that the investigated extracts inhibit one or more of these proteins and induce the retention of DOX. Furthermore, this may be a mechanism underlying the activation of apoptosis and increase in the subG1 phase cell numbers after the DOX/extract treatment [33]. Our further studies should clarify the mechanisms of DOX retention.

The inhibitors of matrix metalloproteinases are considered potential novel agents able to inhibit tumor growth and metastases, but they were shown to be unsuccessful in several clinical trials, which may result from the dual role of matrix metalloproteinases during cancer cell invasion and metastases [34]. These enzymes can degrade extracellular matrix as well as promote cancer cell invasion, migration, and neovascularization [35], but, on the other hand, they are able to reduce cancer growth and vascularization by inducing the generation of angiogenesis inhibitors (angiostatin and tumstatin) [35,36,37,38]. Scientists [39] concluded that MMP-2 and MMP-9 drive metastatic pathways, migration, viability, and secretion of angiogenic factors in two cell lines representing the metastatic and nonmetastatic forms of retinoblastoma cells. The observed inhibition of MMP9 expression in cells treated with the extracts was maintained even after combining these extracts with DOX. Chen et al. [34] showed that the increase in plasma levels of MMP9 promotes tumorigenicity in vivo, and these tumors are smaller and less vascularized compared with those grown in mice with lower MMP9 levels, which is explained by MMP9-induced angiostatin synthesis. MMP9 inhibitors lead to a decrease in the number of tumor colonies, but tumors in vivo are larger and more vascularized, which may provide a rationale for the coadministration of MMP inhibitors and antiangiogenic agents [34]. Our previous study [14] showed good antiangiogenic potential of *M. aquifolium* extracts, especially MAE, and this effect may indicate that these agents are suitable for overcoming the aforementioned issue of tumor vascularization and growth at lower MMP9 levels, while lower MMP9 expression in cells treated with extracts may reduce the metastatic potential of cancer cells. However, further experiments with a broader range of extract doses must be carried out to establish the proapoptotic activity of M. *aquifolium*.

Berberine, the main plant alkaloid of the genus *Mahonia* and the constituent of both *M. aquifolium* extracts involved [14], exhibits antimetastatic potential as well by blocking Wnt/β-catenin signaling pathway [40,41,42]. Furthermore, berberine can activate ZO-1 (Zonula Occludance-1), which participates in the formation of cell-tight junctions and indirectly reduces cell mobility [43]. We have investigated the influence of DOX, extracts, and their combinations on the expression of genes that participate in cell adherence and tight junction formation. *CTNNB1* encodes β-catenin, while OCLN encodes the occludin protein, one of the main components of tight junctions. Taken together, the results obtained here suggest that a decrease in *MMP9* and an increase in *OCLN* expression levels following treatment with a combination of DOX and the investigated extracts may lead to inhibition of cell migration and reduction of the metastatic potential of the treated cells. We have also examined the effects of plant extracts on cell migration, showing that both extracts, alone or in combination with DOX, inhibit cell migration, unlike DOX alone. Colony formation analysis results support our observations that the investigated plant extracts work together with DOX, enhancing its anticancer effects.

## 4. Materials and Methods

### 4.1. Plant Extracts/LC-MC Analyses

The stem bark of cultivated *Mahonia aquifolium* (Pursh) Nutt. was collected in the National Garden park in Pančevo, Serbia, in October 2014. The voucher specimen is deposited at the herbarium of the Institute for Medicinal Plants Research “Dr Josif Pančić “, Belgrade (No. 046/14).

Both dry extracts were obtained from air-dried and finely powdered stem bark of cultivated *M. aquifolium*. MAE was extracted with 70% EtOH at room temperature for 24 h (1:5, *w/v*), while MAW was prepared by ultrasound-assisted extraction with water (1:10, *w/v*) for 30 min. Dry extracts were analyzed by the LC/MS method on Agilent 1200 Series, Agilent Technologies, with a DAD detector on the column Zorbax Eclipse XDB-C18 (RRHT, 50 × 4.6 mm i.d.; 1.8 μm) in combination with 6210 time-of-flight LC/MS system (Agilent Technologies) [14].

### 4.2. Reagents

High-capacity cDNA reverse transcription kit was obtained from Thermo Fisher Scientific (Waltham, MA, USA). All other reagents were purchased from Sigma-Aldrich (St. Louis, MO, USA), unless otherwise specified.

### 4.3. Cell Lines

Human lung adenocarcinoma (A549) cells (ATCC, Manassas, VA, USA) were maintained in RPMI-1640 medium at 37 °C in humidified atmosphere with 5% CO_2_ [44].

### 4.4. MTT Assay

Stock solutions of MAW, MAE, and DOX were dissolved in dimethyl sulfoxide (DMSO) at the concentration of 50 mg/mL (extracts) or 1 mM (DOX). Cells were seeded into 96-well microtiter plates at a density of 5000 cells/well. After 24 h, they were treated with five different extract concentrations (12.5, 25, 50, 100, and 200 μg/mL) or DOX (0.31, 0.62, 1.25, 2.5, and 5 µM). To determine the combined effect of DOX and extracts, cells were treated with various concentrations of DOX in the presence of subtoxic concentrations of MAW or MAE (5, 10, or 20 μg/mL). The control cells were grown in culture medium only. After an additional 72 h of incubation, cell survival was determined by MTT test, as described elsewhere [44,45,46]. The absorbance was measured at 570 nm using Multiskan EX reader (Thermo Labsystems Beverly, MA, USA). The experiments were performed in triplicate, and the data are presented as mean ± standard deviation (SD) of the results obtained in three independent experiments.

Combination index (CI) was used to determine the degree of interaction between DOX and *M. aquifolium*, and its formula is the sum of the ratio of the dose of each drug in the compound to the dose when used alone when the combination and compound produce 50% efficacy [47].

The CI values represent the mean of three experiments with the following values: CI 1.3: antagonism; CI 1.1–1.3: moderate antagonism; CI 0.9–1.1: additive effect; CI 0.8–0.9: slight synergism; CI 0.6–0.8: moderate synergism; CI 0.4–0.6: synergism; and CI 0.2–0.4: strong synergism [48,49].

### 4.5. Cell Cycle Analysis

A549 cells were seeded into six-well plates (2 × 10^5^ cells/well). After 24 h, the cells were treated with concentrations corresponding to IC_20_ or IC_50_ values of DOX with 20 μg/mL of extracts or with combination of IC_20_ or IC_50_ DOX with 20 μg/mL of extracts. The cells were incubated, collected, and fixed. Afterward, the cells were washed, treated with RNase A, stained with propidium iodide, and analyzed using FACSCalibur flow cytometer (BD Biosciences Franklin Lakes, NJ, USA) and CELLQuest software (BD Biosciences) [44]. The obtained results are presented as mean ± SD of the results obtained in three independent experiments.

### 4.6. Cellular Uptake and Retention of Doxorubicin

A549 cells were seeded in six-well plates (2 × 10^5^ cells/well). For DOX uptake experiments, the cells were treated with IC_50_ DOX alone or in combination with 40 μg/mL of extracts. After 24 h, the cells were washed with the medium and analyzed using FACSCalibur. For DOX retention, the cells were treated with IC_50_ of DOX for 24 h, and afterward, the samples were washed and treated with 40 μg/mL of extracts or with the nutrient medium only. After 24 h, the treated cells were washed again and analyzed. Fluorescence intensity measured in cells treated with DOX alone was used as a mark of 100% DOX retention/uptake. All data are presented as mean ± SD obtained in three independent experiments.

### 4.7. Scratch Assay

A549 cells were seeded in 24-well plates (7 × 10^4^ cells/well), where they formed confluent monolayers after 24 h. The monolayers were scraped with a 200 µL pipette tip, and straight, cell-free gaps in the middle of cell monolayers were created. The cells were subsequently washed with nutrient medium and treated with IC_20_ concentration of DOX in the presence or absence of the subtoxic extract concentration (20 µg/mL) or with 20 µg/mL extracts only. The control cells were maintained in nutrient medium only. Images were obtained immediately after making the scratches and after 24 and 48 h of incubation. Three independent experiments were performed. Three representative points were selected in each image; the widths of the gap were measured and averaged. The average gap width at 0 h was considered 100%, and other average gap widths (%) were calculated relative to this value.

### 4.8. Colony Formation Assay

A549 cells (5 × 10^4^ cells/well) were seeded into six-well plates and incubated for 24 h. Subsequently, they were treated with 0.11 µg/mL (IC_20_) of DOX alone, subtoxic concentrations of MAE or MAW (20 µg/mL), or their combination. After an additional 24 h of treatment, the medium was removed from the wells, cells were harvested by trypsinization and replated at different densities, and they were left to grow for 1 week. Afterward, the colonies were stained, and the total number of colonies per well was counted. The experiments were performed in triplicate and repeated three times. For each treatment, the obtained number of colonies/well was divided by the cell-seeding density, and these results were averaged.

### 4.9. Gene Expression Analyses

A549 cells were seeded in 75 cm^2^ cell culture flasks (3 × 10^6^ cells/flasks) and grown for 24 h. Afterward, the cells were treated with subtoxic IC_20_ concentrations of DOX, 20 µg/mL extracts, or combination of IC_20_ DOX and 20 µg/mL extracts for 24 h. The control cells were grown in the culture medium. Following incubation, cells were collected by trypsinization, washed using PBS, and the collected cell pellet was stored at −80 °C until further experiments.

Total RNA was extracted from cells using TRI Reagent BD kit in line with the manufacturer’s recommendations. RNA bands were visualized on a UV transilluminator, and RNA concentration was determined spectrophotometrically (BioSpec Nano, Shimadzu, Kyoto, Japan). Primary cDNA was prepared with RT-PCR using random primers, and 2 µg of total RNA was used as a template for MultiScribe reverse transcriptase in a high-capacity cDNA reverse transcription kit in line with the manufacturer’s instructions.

### 4.10. Real-Time PCR Amplification

All target transcripts were detected using quantitative real-time PCR (qPCR) and TaqMan assays. TaqMan gene expression assays (*OCLN*: Hs00170162_m1, *CTNNB1*: Hs00355049_m1, *MMP2*: Hs01548727_m1, *MMP9*: Hs00234579_m1, and *ERCC1*: Hs01012158_m1) contained 20× mix of unlabeled PCR primers and TaqMan (Minor groove blinder (MGB)) probes (FAM dye-labeled). Glyceraldehyde-3-phosphate dehydrogenase (*GAPDH*) levels obtained using TaqMan control reagents (Hs02758991_g1) were used as an endogenous control. PCR reactions were performed using an ABI Prism 7500 sequence detection system (Applied Biosystems) under previously described conditions [50].

### 4.11. Statistical Analysis

One-way ANOVA with Tukey’s multiple comparison test was used for multiple data comparisons and *p* value < 0.05 was considered statistically significant.

## 5. Conclusions

Taken together, our results show that *M. aquifolium* extracts and their active principles should be investigated further, either on their own or in combination with other anticancer drugs similar to DOX. It remains to be determined whether changes in MMP9 expression and DOX retention underlie these effects and which alkaloids are responsible for the anticancer activities against A549 cells. In addition, the synergistic effects of extracts with DOX in other systems remain to be confirmed.

## Figures and Tables

**Figure 1 molecules-25-05233-f001:**
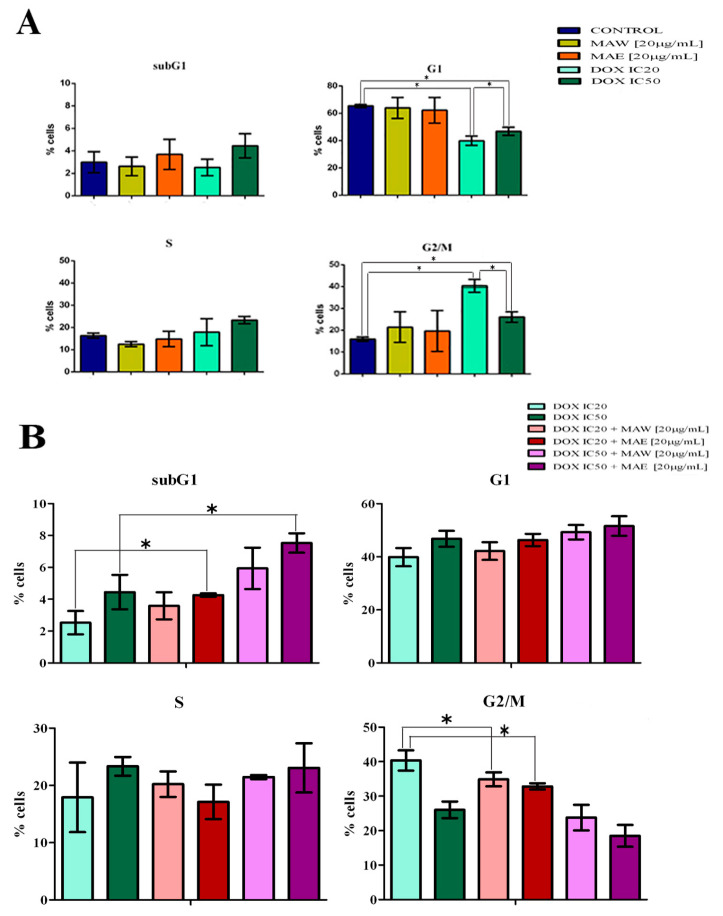
Changes in the cell cycle phase distribution of A549 cells after 24 h of treatment induced by (**A**) MAW (20 μg/mL), MAE (20 μg/mL), DOX IC20, or DOX IC50 compared to the control and (**B**) combination DOX/extract treatment compared to DOX treatment alone.

**Figure 2 molecules-25-05233-f002:**
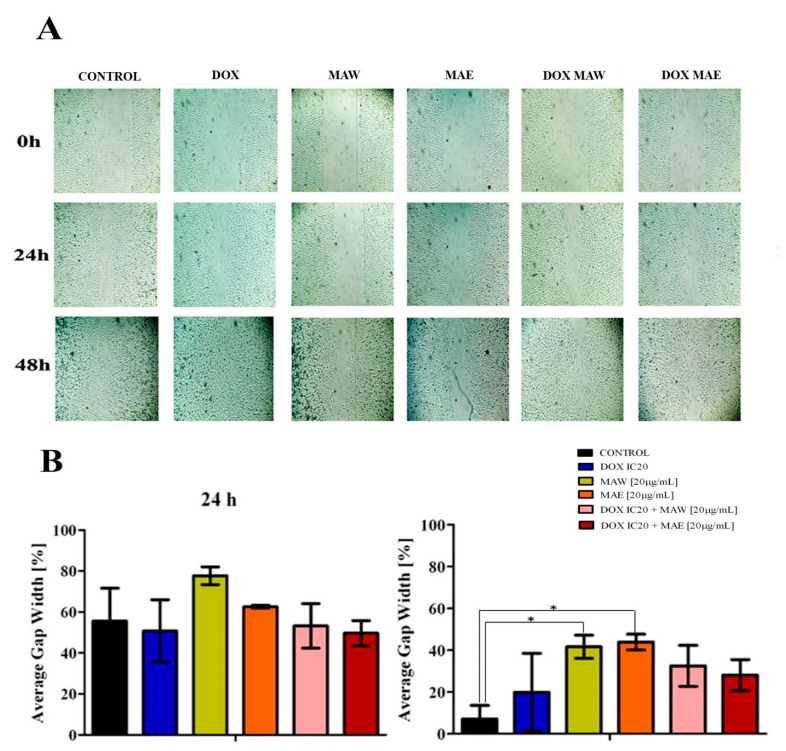
Effects of *M. aquifolium* extracts and DOX on migration of A549 cells. The cells were treated with DOX IC20, MAW (20 μg/mL), MAE (20 μg/mL), or their combinations. (**A**) Representative images of one of three independent experiments. (**B**) Quantitative analysis of results presented in (A).

**Figure 3 molecules-25-05233-f003:**
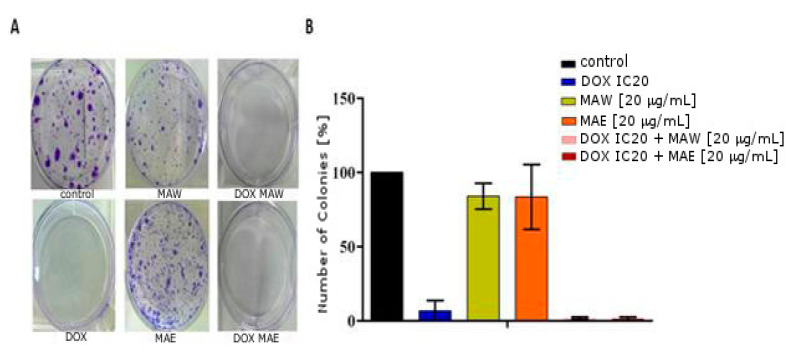
Effects of *M. aquifolium* extracts and DOX on A549 colony forming ability. (**A**) Representative images of colonies stained with Coomassie Brilliant Blue following treatment with DOX IC20, MAW (20 μg/mL), MAE (20 μg/mL), or their combinations. Experiments were performed at least three times. (**B**) Quantitative analysis of results presented in (A).

**Figure 4 molecules-25-05233-f004:**
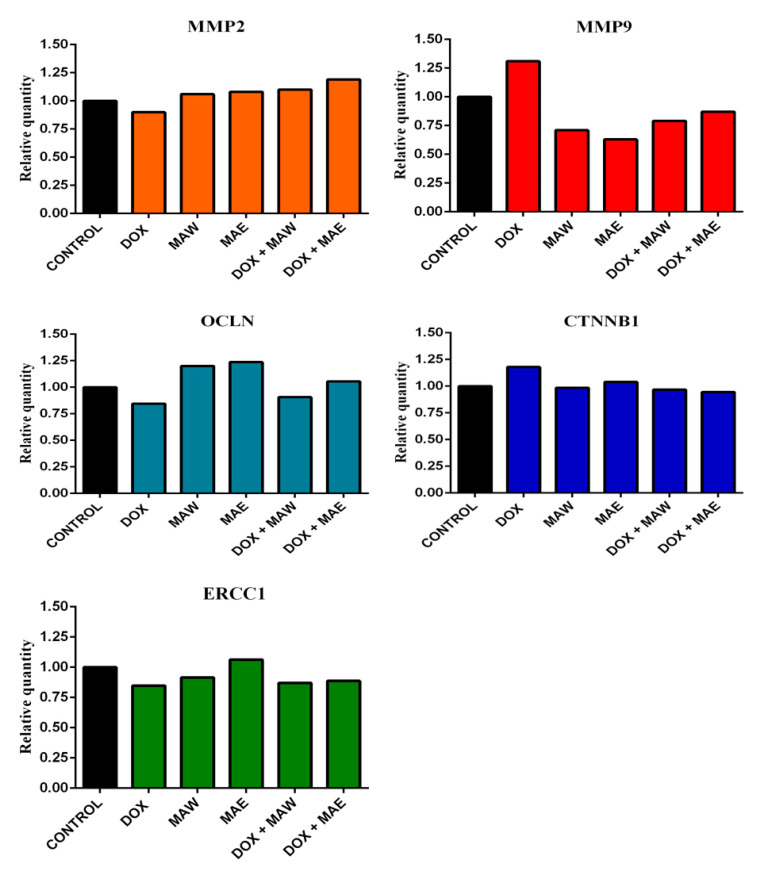
Effects of DOX IC20, MAE (20 μg/mL), MAW (20 μg/mL), or their combinations on gene expression.

**Table 1 molecules-25-05233-t001:** Concentrations of doxorubicin alone or in combination with extracts that induced 50% decrease in cell survival after 72 h of treatment.

	IC50 (µg/mL)
	A549
**DOX**	0.4457 ± 0.0154
DOX + 5 µg/mL MAW	0.0234 ± 0.0022
DOX + 10 µg/mL MAW	0.0141 ± 0.0046
DOX + 20 µg/mL MAW	0.0067 ± 0.0018
DOX + 5 µg/mL MAE	0.0113 ± 0.0006
DOX + 10 µg/mL MAE	0.0103 ± 0.0016
DOX + 20 µg/mL MAE	0.0036 ± 0.0012

IC_50_ values are presented as the mean ± standard deviation (SD) from three independent experiments; DOX: doxorubicin; MAW: water extract of *M. aquifolium*; MAE: ethanol extract of *M. aquifolium*.

**Table 2 molecules-25-05233-t002:** Isobolographic analysis.

	CI
**Treatment**	A549
DOX + 5 µg/mL MAW	0.14
DOX + 10 µg/mL MAW	0.21
DOX + 20 µg/mL MAW	0.38
DOX + 5 µg/mL MAE	0.12
DOX + 10 µg/mL MAE	0.22
DOX + 20 µg/mL MAE	0.40

CI: combination index.

**Table 3 molecules-25-05233-t003:** Effects of *M. aquifolium* water and ethanol extracts on the uptake and retention of doxorubicin in A549 cells.

Treatment	Uptake	Retention
DOX IC_50_	100	100
DOX IC_50_ + MAW (40 μg/mL)	92.34 ± 6.64	112.97 ± 0.25
DOX IC_50_ + MAE (40 μg/mL)	96.58 ± 2.77	122.23 ± 1.60

The results are presented as the mean ± standard deviation (SD) from three independent experiments.

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
