# Peer review of "Mahonia aquifolium Extracts Promote Doxorubicin Effects against Lung Adenocarcinoma Cells In Vitro"

_molecules, 2020, doi:10.3390/molecules25225233_

Round 1
Reviewer 1 Report
Dear Authors,
The present study ID: molecules-935055 entitled " Mahonia aquifolium extracts promote doxorubicin effects against lung adenocarcinoma cells in vitro" written by authors Ana Damjanovic, Branka Kolundžija, Ivana Z. Matic, Ana Krivokuca, Gordana Zdunic, Katarina Savikin, Radmila Jankovic, Jelena Antić Stankovic, and Tatjana P. Stanojkovic is devoted to information on Mahonia aquifolium extracts promote doxorubicin effects.
The study provides interesting information on the effect of natural plant extract on the effect of doxorubicin in terms of cytotoxicity. Some anti-cancer activity has been confirmed in Mahonia aquifolium in the past, so its use in combination with doxorubicin is relevant. I evaluate very positively the number of adequate methods used to evaluate the effect of the monitored matrix in the study.
The text is written clearly and in comprehensive style, divided into suitable chapters.
The results are interpreted in an appropriate way through tables and graphs. The text is written in essentially the correct English language. I have no fundamental reproaches to study.
Minor comment:
Page 1.: I recommended to add some other keywords in order to specify the topic of MS in better way (eg. "cytotoxicity", "adenocarcinoma", etc.).
Page 2.: The last sentence of 2.1. chapter (page No. 2) - Must be moved between results and discussion.
Page 4.: In chapter 2.9 a different font/font size is used, it is necessary to edit. Page 11.: "Conclusion" - should be written more specific.
Author Response
Dear colleague,
Thanks for your comments. We accepted all suggestions.

Reviewer 2 Report
In the manuscript entitled „Mahonia aquifolium extracts promote doxorubicin effects against lung adenocarcinowma cells in vitro” by Ana Damjanović et al. cytotoxic effects of doxorubicin combined with Mahonia aquifolium water and ethanol extracts are reported. Furthermore, the authors present their investigations of the effect of doxorubicin and M. aquifolium extracts on cell proliferation, cell migration, colony formation and the expression of genes involved in the regulation of cancer progression, neovascularization and excision repair.
The rationale of the study is well formulated and the obtained results are convincing. To confirm the synergistic effect of M. aquifolium extracts with doxorubicin, isobologram method was used. Enhancing the cytotoxicity of doxorubicin against cancer cells by combination therapy with phytochemicals will allow to diminish therapeutic doses of doxorubicin reducing the side effects of this anticancer agent.
The manuscript is written in correct English language. The discussion of the obtained results seems to me enough comprehensive. However, the part concerning apoptosis is too speculative. To investigate proapoptotic activity of M. aquifolium and doxorubicin/extracts combinations further experiments should be carried out with broader range of extract doses. The role of the enzymes MMP2 and MMP9 in cellular migration and angiogenesis was also reported by Webb et al.: BMC Cancer 17: 434, 2017.
My objection concerns the introduction to isobolographic analysis for which the basics could be better explained. I am not sure that the cited article of Former et al., 2012, is the right one (see: Gessner PK. Toxicology 105, 161-179, 1995 or more recently Huang R, Frontiers in Pharmacology, 29 October 2019, open access)
In the section Conclusion, it should be mentioned that the studies were performed on a single cancer cell line A 549. In my opinion, the generalization to all cancer cells is premature.
The references should be prepared according to the MDPI Author Instructions for the journal „Molecules”.
Minor tips:
- Page 3, the last paragraph of 2.4. MTT assay; CI values are interpreter as follows: CI>1.3; antagonism; 1.1<CI<1.3; moderate antagonism, and so on…
- Page 3, 4 line from below: All data are presented…., please reformulate the sentence.
- Page 9, 4 line from below: from the our extracts.
- In the figures 1 and 2, asterisks (*) are not explained in figure captions.
- Page 11, line 10: increase of the plasma level.
- In the references, the position 23 (Kumar A et al., 2015): the second author is Chopra E. K., Ekavali is his first name.
Author Response
Dear colleague,
Thanks for the comments and suggestions.
1. We added data on the role of enzymes MMP2 and MMP9 in cell migration and angiogenesis
2. We explained the calculation of the CI index.
3. We have improved the conclusion

Reviewer 3 Report
As a general comment, the study is mainly descriptive, without any insight in the possible mechanism of action of the extract – if a possible mechanism of action of an extract can be identified for a mixture of molecules.
- Authors should specify why they have chosen a lung adenocarcinoma cell line for their study: is there any reason that make a lung cancer more sensitive to the effects of M. aquifolium than others? Is there any evidence that molecules present in this extract can actually reach this target in the body?
- If I understood correctly, the extracts have been prepared once, aliquoted and then used for all the experiments. Which is the variability in the concentrations of active compounds among different extractions?
- It is not clear why the authors did not test berberine alone in the experiments, as positive control.
-In many cases, the effects of MAW and MAE extract are identical or not statistically different (that is, they are the same…), eve inf the content of berberine is roughly double in MAE. How do the authors justify this difference? If the effects were due to berberine, what observed with MAW 2X should be similar or identical to MAE 1X, but this is not the case in many situations.
-The conclusions should be mitigated, as the “beneficial effects” have been seen only in one case, and we actually do not know if those concentrations of extracts can be reached in the human body, without having an impact or collateral effects on normal cells, tissues, organs.
Author Response
Dear colleague,
Thanks for the comments and suggestions.
1.As a general comment, the study is mainly descriptive, without any insight in the possible mechanism of action of the extract – if a possible mechanism of action of an extract can be identified for a mixture of molecules.
You are right when you say that for an extract composed of several active principles is difficult to examine the mechanisms of activity, since each component may have a different mechanism of action. We considered that is necessary to examine the potential activities of these extracts, then to identify the active principles (ref 1), and than in our next research we plan to examine the molecular mechanisms of the identified active principles.
1.Dejan Gođevac, Ana Damjanović, Tatjana P Stanojković, Boban Anđelković, Gordana Zdunić. Identification of cytotoxic metabolites from Mahonia aquifolium using 1 H NMR-based metabolomics approach. J Pharm Biomed Anal. 2018 Feb.
2.Authors should specify why they have chosen a lung adenocarcinoma cell line for their study: is there any reason that makes a lung cancer more sensitive to the effects of M. aquifolium than others? Is there any evidence that molecules present in this extract can actually reach this target in the body?
The cytotoxic activity of the extracts was examined against a panel of malignant cell lines. A549 cells were the most invasive and also the least sensitive to doxorubicin. Furthermore, doxorubicin is the part of medical therapy in the treatment of lung cancer. There are numerous studies examining the anticancer potential of berberine-type alkaloids in vivo. Previous research showed that both berberine and berbamine can inhibit the growth of lung cancer cells in in vivo systems (ref 2, 3), and based on these results we can conclude that the active principles of our extracts have potential to reach this target in the body.
2.Xu J, Long Y, Ni L, et al. Anticancer effect of berberine based on experimental animal models of various cancers: a systematic review and meta-analysis. BMC Cancer. 2019;19(1):589. Published 2019 Jun 17. doi:10.1186/s12885-019-5791-1
3. Hou ZB, Lu KJ, Wu XL, Chen C, Huang XE, Yin HT. In vitro and in vivo antitumor evaluation of berbamine for lung cancer treatment. Asian Pac J Cancer Prev. 2014;15(4):1767-9. doi: 10.7314/apjcp.2014.15.4.1767. PMID: 24641406.
3.If I understood correctly, the extracts have been prepared once, aliquoted and then used for all the experiments. Which is the variability in the concentrations of active compounds among different extractions?
After extraction, the extracts were evaporated, and they were solid. The both extracts then were dissolved in DMSO (extract stock). The required volumes were taken from that stock to make appropriate dilutions with the medium. Since both solid extracts were completely soluble in DMSO, we can say that dilutions were always made from the same initial stock concentration. This is a standardized procedure and there should be no deviations in the desired concentrations during the study.
4. It is not clear why the authors did not test berberine alone in the experiments, as positive control.
This is a very good question. The anti-cancer potential of berberine has been tested and proven. Our research showed that extracts, except berberine, have other alkaloids that are responsible for activity. In our published work (ref 1) we showed that palmatine and berbamine are even more than berberine responsible for the cytotoxic activity. All these alkaloids together have potential for the synergistic and/or additive effects. At this stage of the research we could not say with certainty which alkaloid or group of alkaloids has the most intense anticancer effect, so we decided not to take a positive control.
5. Conclusions
Taken together, our results demonstrate that M. aquifolium extracts with their active principles have the potential to be further investigated alone or in combination with other anticancer drugs, such as DOX. It remains to be elucidated whether the changes in MMP9 expression and DOX retention underlie these effects, which alkaloids are responsible for the anticancer activities, and to confirm the synergistic effects with DOX in more complex systems.
